# Distributed Guaranteed-Performance Consensus of Networked Systems Without Involving the Feasibility Condition: A Hierarchical Algorithm

Lei Chen
*School of Automation Engineering*
*University of Electronic Science and Technology of China*
Chengdu, China
abc278559033@163.com

*Abstract*—This paper utilizes a hierarchical algorithm to focus on the distributed guaranteed-performance consensus problem of multiagent systems. With this algorithm, the consensus problem can be transformed into the tracking problem between agents in adjacent layers. Compared to the existing hierarchical algorithm, the proposed hierarchical rules consider the practical significance of topological weights, which makes the partition results more reasonable. In addition, a shifting function is designed and incorporated into the conventional prescribed performance control approach, which eliminates the existing feasibility condition. Meanwhile, the information concerning the Laplacian matrix is no longer required when prescribing the bounds of the tracking errors. Based on Lyapunov theory, the sufficient conditions for the boundedness of all signals in the closed-loop system are derived. Finally, two simulation examples verify the efficacy of the scheme.

*Index Terms*—Distributed consensus, hierarchical algorithm, multiagent systems, prescribed performance control.

## I. INTRODUCTION

**T**HE cooperative control technology of multiagent systems (MASs) has been widely devoted to numerous fields in practice, thus attracting significant concentration from individuals. As one of the hot investigation issues of cooperative control, consensus control demands that a group of agents' cluster behaviour remain consistent during their movement procedure. In particular, when a leader exists in this group of agents, the related problem is named the consensus tracking problem, and considerable works [1]– [4] have investigated this problem under multiple cases. For instance, recognizing that the agents' network bandwidth in practice is limited due to the restrictions of hardware devices, the authors in [1] designed an adaptive event-triggered mechanism in the development of a consensus tracking algorithm for MASs, achieving the desired consensus mission while decreasing the consumption of unnecessary network resources. Moreover, the relationships of collaboration and competition coexist among agents. Following this reality, Niu *et al*. [4] focused on the bipartite consensus tracking (BCT) problem of the signed-directed networks.

Currently, the distributed control approach that utilizes agents' local information or their neighbors' information is one of the effective ways to solve the consensus tracking control (CTC) problem of MASs. This approach enables the agents to achieve parallel computing and division of labor and improves the response speed and execution efficiency of systems. In addition, distributed control allows systems to be more flexible, and the agents can make autonomous decisions and adjustments according to the environment changes and task requirements. So far, numerous distributed control methods have been proposed in [6]– [12]. For example, Zhao *et al*. [6] designed a distributed control strategy, achieving the consensus tracking task of a set of heterogeneous MASs consisting of first- and second-order dynamics, where each agent is subjected to Bouc-Wen hysteresis input. In [12], the authors utilized a hierarchical algorithm to divide the agents into different layers, and then they researched the distributed BCT problem of signed directed networks with unbalanced structures. Although this hierarchical algorithm is helpful for the design of distributed control schemes, it does not consider the practical meaning of the topology weights, such that the rationality of the partition results needs to be improved when dealing with some complex topologies. How to solve this problem is one of the driving forces of this paper.

It should be emphasized that the works [12] did not analyze the consensus tracking problem of MASs from the perspective of tracking performance. Some practical control missions require agents to track reference signals with a desired precision within a specified time. Especially for agents in uncertain environments, maintaining satisfactory tracking performance is not an easy task. Therefore, the literature [13] adopt the prescribed performance control (PPC) method to maintain that the tracking performance does not decrease during the motion of the controlled systems. For instance, via a designed observer, Zhang *et al*. [13] focused on the guaranteed-performance CTC problem of MASs with unmeasured states. In [1], a finite-time consensus tracking approach for MASs is proposed based on PPC technique, in which the designer can directly set the settling time of tracking errors. Although the conventional PPC technique can improve the tracking performance of MASs to a certain extent, this technique needs the tracking error to satisfy a feasibility condition (the initial value of the

tracking error must be restricted to the range enveloped by the performance function). In addition, this approach relies on the Laplacian matrix to prescribe the bound of tracking error, which induces some inconvenience in implementing the control scheme. Therefore, how to solve the above deficiencies is the primary research motivation of this paper.

Inspired by the observations above, this paper investigates the distributed guaranteed-performance CTC problem of MASs based on a hierarchical algorithm. The main contributions can be given as follows.

1) A shifting function is designed to remove the feasibility condition in the conventional PPC approach.
2) Different from the CTC schemes [1] where the tracking precision depends on the Laplacian matrix, the proposed scheme can preset the bounds of tracking errors independently of the Laplacian matrix.
3) Compared with the hierarchical algorithm in [13], new hierarchical rules are designed based on the practical significance of topological weights, which makes the hierarchical results more reasonable.

## II. THE MAIN RESULTS

### A. Error Transformation

In this section, an error transformation mechanism is presented to guarantee that the second control objective is completed. The bipartite distributed error $s_{i1}$ is provided as

$$s_{i1} = \sum_{\zeta=1}^{M} |a_{i\zeta}|(y_i - \text{sgn}(a_{i\zeta})y_\zeta) + |b_i|(y_i - \text{sgn}(b_i)y_d) \quad (1)$$

where $\text{sgn}(\cdot)$ denotes the symbolic function. Based on $s_{i1}$, we present the following error transformation mechanism

$$s_{i1} = \vartheta(t)\rho(z_{i1}) \quad (2)$$

where $\rho(z_{i1}) = \frac{2}{\pi}\arctan(z_{i1})$. Define $c_i = \sum_{\zeta=1}^{M} |a_{i\zeta}| + |b_i|$, then we can further get

$$dz_{i1} = \sigma_i\Big(c_i(x_{i2} + g_{i1}) - \sum_{\zeta=1}^{M} a_{i\zeta}(x_{\zeta 2} + g_{\zeta 1}) - b_i\dot{y}_l - o_i\Big)dt$$

$$+ \sigma_i\Big(c_i h_{i1} - \sum_{\zeta=1}^{M} a_{i\zeta}h_{\zeta 1}\Big)d\omega \quad (3)$$

where $o_i = \frac{2}{\pi}\arctan(z_{i1})\dot{\vartheta}(t)$ and $\sigma_i = \frac{\pi(1+z_{i1}^2)}{2\vartheta(t)}$.

### B. Controller Design

This section gives the design process of the controller. We give the following coordinate transformation mechanism.

$$\begin{cases} \lambda_{i1} = z_{i1} \\ \lambda_{ij} = x_{ij} - \bar{\alpha}_{i(j-1)}, \quad j = 2, 3, \dots n_i \end{cases} \quad (4)$$

where $\bar{\alpha}_{i(j-1)}$ denotes the output of the first-order filter concerning the virtual controller $\alpha_{i(j-1)}$.

The compensating signal $\xi_{i1}$ is designed as

$$\dot{\xi}_{i1} = -(l_{i1} + 1)\xi_{i1} + \sigma_i c_i(\bar{\alpha}_{i1} - \alpha_{i1}) + \sigma_i c_i \xi_{i2}$$
$$- k_{i1}\sigma_i c_i \text{sgn}(\xi_{i1}) \quad (5)$$

where $l_{i1}$ and $k_{i1}$ denote positive designed parameters.

To solve unknown functions $G_{i1}$ and $\mathcal{H}_i\mathcal{H}_i^T$, NNs $\phi_{i11}^T\Lambda_{i11}(X_{i11})$ and $\phi_{i12}^T\Lambda_{i12}(X_{i12})$ are adopted, respectively. For any given constants $\delta_{i11}^* > 0$ and $\delta_{i12}^* > 0$, the following relationships hold

$$\begin{cases} G_{i1} = \phi_{i11}^T\Lambda_{i11}(X_{i11}) + \delta_{i11}(X_{i11}) \\ \mathcal{H}_i\mathcal{H}_i^T = \phi_{i12}^T\Lambda_{i12}(X_{i12}) + \delta_{i12}(X_{i12}) \end{cases}$$

where $\delta_{i11}(X_{i11})$ and $\delta_{i12}(X_{i12})$ denote approximation errors. The virtual controller $\alpha_{i1}$ is designed as

$$\alpha_{i1} = -\frac{l_{i1} + 1}{\sigma_i c_i}\lambda_{i1} + \frac{1}{c_i}\Big(b_i\dot{y}_l + \frac{2}{\pi}\arctan(z_{i1})\dot{\vartheta}(t)$$

$$- \frac{3\epsilon_{i11}^{\frac{4}{3}}}{4}\bar{\lambda}_{i1}\sigma_i^{\frac{1}{3}}\hat{\Psi}_i\|\Lambda_{i11}\|^{\frac{4}{3}} - \frac{3\epsilon_{i13}^2}{4}\bar{\lambda}_{i1}\sigma_i^3\hat{\Psi}_i\|\Lambda_{i12}\|^2$$

$$- \frac{3}{4}\bar{\lambda}_{i1}\sigma_i^{\frac{1}{3}} - \frac{3\epsilon_{i12}^{\frac{4}{3}}}{4}\bar{\lambda}_{i1}\sigma_i^{\frac{1}{3}} - \frac{3\epsilon_{i14}^2}{4}\bar{\lambda}_{i1}\sigma_i^3\Big) \quad (6)$$

Then, $\mathcal{L}V_{i1}$ can be further rewritten as

$$\mathcal{L}V_{i1} \leq \bar{\lambda}_{i1}^3\sigma_i c_i\bar{\lambda}_{i2} - (l_{i1} + 1)\bar{\lambda}_{i1}^4 + \frac{1}{\varphi_i}\tilde{\Psi}_i(\Gamma_{i1} - \dot{\hat{\Psi}}_i) + \gamma_{i1}$$

where $\Gamma_{i1} = \frac{3\varphi_i\epsilon_{i11}^{\frac{4}{3}}}{4}\bar{\lambda}_{i1}^4\sigma_i^{\frac{4}{3}}\|\Lambda_{i11}\|^{\frac{4}{3}} + \frac{3\varphi_i\epsilon_{i13}^2}{4}\bar{\lambda}_{i1}^4\sigma_i^4\|\Lambda_{i12}\|^2$ and $\gamma_{i1} = \frac{k_{i1}^4 c_i^4}{4} + \frac{1}{4\epsilon_{i11}^4} + \frac{\delta_{i11}^{*4}}{4\epsilon_{i12}^4} + \frac{3}{4\epsilon_{i13}^2} + \frac{3\delta_{i12}^{*2}}{4\epsilon_{i14}^2}$.

**Step $\tau$** ($2 \leq \tau < n_i$): Construct Lyapunov function as

$$V_{i\tau} = V_{i(\tau-1)} + \frac{1}{4}\bar{\lambda}_{i\tau}^4$$

where $\bar{\lambda}_{i\tau} = \lambda_{i\tau} - \xi_{i\tau}$ represents the compensated error, and the compensating signal $\xi_{i\tau}$ is designed as

$$\dot{\xi}_{i\tau} = -(l_{i\tau} + 1)\xi_{i\tau} + (\bar{\alpha}_{i\tau} - \alpha_{i\tau}) - \check{\varepsilon}\xi_{i(\tau-1)}$$
$$+ \xi_{i(\tau+1)} - k_{i\tau}\text{sgn}(\xi_{i\tau}) \quad (7)$$

where $l_{i\tau} > 0$ and $k_{i\tau} > 0$ are designed parameters. $\check{\varepsilon} = \sigma_i c_i$ if and only if $\tau = 2$. Otherwise, $\check{\varepsilon} = 1$. Then, the infinitesimal generator of $V_{i\tau}$ can be calculated as

$$\mathcal{L}V_{i\tau} \leq \bar{\lambda}_{i\tau}^3\Big[\bar{\lambda}_{i(\tau+1)} + \alpha_{i\tau} + g_{i\tau} - \dot{\bar{\alpha}}_{i(\tau-1)} + (l_{i\tau} + 1)\xi_{i\tau}$$

$$+ \check{\varepsilon}\xi_{i(\tau-1)} + k_{i\tau}\text{sgn}(\xi_{i\tau})\Big] + \check{\varepsilon}\bar{\lambda}_{i(\tau-1)}^3\bar{\lambda}_{i\tau}$$

$$- \sum_{j=1}^{\tau-1} l_{ij}\bar{\lambda}_{ij}^4 - \bar{\lambda}_{i(\tau-1)}^4 + \frac{\tilde{\Psi}_i}{\varphi_i}\Big(\Gamma_{i(\tau-1)} - \dot{\hat{\Psi}}_i\Big)$$

$$+ \gamma_{i(\tau-1)} + \frac{3}{2}\bar{\lambda}_{i\tau}^2 h_{i\tau} h_{i\tau}^T$$

For unknown nonlinear function $g_{i\tau}$, we have

$$\bar{\lambda}_{i\tau}^3 g_{i\tau} \leq \frac{3\epsilon_{i\tau 1}^{\frac{4}{3}}}{4}\bar{\lambda}_{i\tau}^4\Psi_i\|\Lambda_{i\tau 1}\|^{\frac{4}{3}} + \frac{1}{4\epsilon_{i\tau 1}^4} + \frac{3\epsilon_{i\tau 2}^{\frac{4}{3}}}{4}\bar{\lambda}_{i\tau}^4 + \frac{\delta_{i\tau 1}^{*4}}{4\epsilon_{i\tau 2}^4}$$

where $\epsilon_{i\tau1} > 0$ and $\epsilon_{i\tau2} > 0$ are designed parameters, and $\delta_{i\tau1}^* > 0$ is any given constant.

$$\frac{3}{2}\bar{\lambda}_{i\tau}^2 h_{i\tau} h_{i\tau}^T \leq \frac{3\epsilon_{i\tau3}^2}{4}\bar{\lambda}_{i\tau}^4 \Psi_i \|\Lambda_{i\tau2}\|^2 + \frac{3}{4\epsilon_{i\tau3}^2}$$
$$+ \frac{3\epsilon_{i\tau4}^2}{4}\bar{\lambda}_{i\tau}^4 + \frac{3\delta_{i\tau2}^{*2}}{4\epsilon_{i\tau4}^2}$$

where $\epsilon_{i\tau3} > 0$ and $\epsilon_{i\tau4} > 0$ are designed parameters and $\delta_{i\tau2}^* > 0$ is any given constant. By Lemma **??**, one gets

$$\begin{cases} \breve{\varepsilon}\bar{\lambda}_{i(\tau-1)}^3 \bar{\lambda}_{i\tau} \leq \bar{\lambda}_{i(\tau-1)}^4 + \frac{3^3}{4^4}\breve{\varepsilon}^4 \bar{\lambda}_{i\tau}^4 \\ \bar{\lambda}_{i\tau}^3 k_{i\tau}\mathrm{sgn}(\xi_{i\tau}) \leq \frac{3}{4}\bar{\lambda}_{i\tau}^4 + \frac{k_{i\tau}^4}{4} \end{cases}$$

The virtual controller $\alpha_{i\tau}$ is designed as

$$\alpha_{i\tau} = -(l_{i\tau}+1)\lambda_{i\tau} + \dot{\bar{\alpha}}_{i(\tau-1)} - \frac{3\epsilon_{i\tau1}^{\frac{4}{3}}}{4}\bar{\lambda}_{i\tau}\hat{\Psi}_i\|\Lambda_{i\tau1}\|^{\frac{4}{3}}$$
$$- \frac{3\epsilon_{i\tau3}^2}{4}\bar{\lambda}_{i\tau}\hat{\Psi}_i\|\Lambda_{i\tau2}\|^2 - \frac{3\epsilon_{i\tau2}^{\frac{4}{3}}}{4}\bar{\lambda}_{i\tau} - \frac{3\epsilon_{i\tau4}^2}{4}\bar{\lambda}_{i\tau}$$
$$- \frac{3}{4}\bar{\lambda}_{i\tau} - \frac{3^3}{4^4}\breve{\varepsilon}^4\bar{\lambda}_{i\tau} - \breve{\varepsilon}\xi_{i(\tau-1)} \tag{8}$$

Then, $\mathcal{L}V_{i\tau}$ satisfy the following inequaltiy

$$\mathcal{L}V_{i\tau} \leq \bar{\lambda}_{i\tau}^3 \bar{\lambda}_{i(\tau+1)} - \sum_{j=1}^{\tau} l_{ij}\bar{\lambda}_{ij}^4 - \bar{\lambda}_{i\tau}^4 + \frac{\tilde{\Psi}_i}{\varphi_i}(\Gamma_{i\tau} - \dot{\hat{\Psi}}_i) + \gamma_{i\tau}$$

where $\Gamma_{i\tau} = \Gamma_{i(\tau-1)} + \frac{3\varphi_i\epsilon_{i\tau1}^{\frac{4}{3}}}{4}\bar{\lambda}_{i\tau}^4\|\Lambda_{i\tau1}\|^{\frac{4}{3}} + \frac{3\varphi_i\epsilon_{i\tau3}^2}{4}\bar{\lambda}_{i\tau}^4\|\Lambda_{i\tau2}\|^2$ and $\gamma_{i\tau} = \gamma_{i(\tau-1)} + \frac{1}{4\epsilon_{i\tau1}^4} + \frac{\delta_{i\tau1}^{*4}}{4\epsilon_{i\tau2}^4} + \frac{3}{4\epsilon_{i\tau3}^2} + \frac{3\delta_{i\tau2}^{*2}}{4\epsilon_{i\tau4}^2} + \frac{k_{i\tau}^4}{4}$.

**Step $n_i$**: We design the compensating signal as

$$\dot{\xi}_{in_i} = -(l_{in_i}+1)\xi_{in_i} - \xi_{i(n_i-1)} - k_{in_i}\mathrm{sgn}(\xi_{in_i})$$

where $l_{in_i} > 0$ and $k_{in_i} > 0$ are designed parameters. Then, select the following Lyapunov function as

$$V_{in_i} = V_{i(n_i-1)} + \frac{1}{4}\bar{\lambda}_{in_i}^4$$

where $\bar{\lambda}_{in_i} = \lambda_{in_i} - \xi_{in_i}$ denotes the compensated error. The infinitesimal generator of $V_{in_i}$ can be calculated as

$$\mathcal{L}V_{in_i} \leq \bar{\lambda}_{i(n_i-1)}^3 \bar{\lambda}_{in_i} - \sum_{j=1}^{n_i-1} l_{ij}\bar{\lambda}_{ij}^4 - \bar{\lambda}_{i(n_i-1)}^4 + \gamma_{i(n_i-1)}$$
$$+ \frac{1}{\varphi_i}\tilde{\Psi}_i\left(\Gamma_{i(n_i-1)} - \dot{\hat{\Psi}}_i\right) + \frac{3}{2}\bar{\lambda}_{in_i}^2 h_{in_i} h_{in_i}^T$$
$$+ \bar{\lambda}_{in_i}^3\left(u_i + g_{in_i} - \dot{\bar{\alpha}}_{i(n_i-1)} + (l_{in_i}+1)\xi_{in_i}\right.$$
$$\left. + \xi_{i(n_i-1)} + k_{in_i}\mathrm{sgn}(\xi_{in_i})\right) \tag{9}$$

Similar to inequalities (**??**) and (**??**), the unknown function $\bar{\lambda}_{in_i}^3 g_{in_i}$ satisfies the following inequality

$$\bar{\lambda}_{in_i}^3 g_{in_i} \leq \frac{3\epsilon_{in_i1}^{\frac{4}{3}}}{4}\bar{\lambda}_{in_i}^4 \Psi_i\|\Lambda_{in_i1}\|^{\frac{4}{3}} + \frac{1}{4\epsilon_{in_i1}^4} + \frac{3\epsilon_{in_i2}^{\frac{4}{3}}}{4}\bar{\lambda}_{in_i}^4 + \frac{\delta_{in_i1}^{*4}}{4\epsilon_{in_i2}^4}$$

where $\epsilon_{in_i1} > 0$ and $\epsilon_{in_i2} > 0$ are designed parameters, and $\delta_{in_i1}^* > 0$ is any given constant. Moreover, we can obatin

$$\frac{3}{2}\bar{\lambda}_{in_i}^2 h_{in_i} h_{in_i}^T \leq \frac{3\epsilon_{in_i3}^2}{4}\bar{\lambda}_{in_i}^4 \Psi_i\|\Lambda_{in_i1}\|^2 + \frac{3}{4\epsilon_{in_i3}^2}$$
$$+ \frac{3\epsilon_{in_i4}^2}{4}\bar{\lambda}_{in_i}^4 + \frac{3\delta_{in_i2}^{*2}}{4\epsilon_{in_i4}^2}$$

where $\epsilon_{in_i3} > 0$ and $\epsilon_{in_i4} > 0$ are designed parameters, and $\delta_{in_i2}^* > 0$ is any given constant. Moreover, we have

$$\begin{cases} \bar{\lambda}_{i(n_i-1)}^3 \bar{\lambda}_{in_i} \leq \bar{\lambda}_{i(n_i-1)}^4 + \frac{3^3}{4^4}\bar{\lambda}_{in_i}^4 \\ \bar{\lambda}_{in_i}^3 k_{in_i}\mathrm{sgn}(\xi_{in_i}) \leq \bar{\lambda}_{in_i}^4 + \frac{3^3}{4^4}k_{in_i}^4 \end{cases}$$

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
