# OpenReview forum: "Distributed Guaranteed-Performance Consensus of Networked Systems Without Involving the Feasibility Condition: A Hierarchical Algorithm"
_IEEE.org/ICIST/2024/Conference — IEEE ICIST 2024 Conference Submission_

### Official Review · Reviewer_LmPm · 2024-08-20
**Review Results**

**Rating:** 5
**Confidence:** 4

**Review:**

The paper presents an innovative approach to the distributed guaranteed-performance consensus problem in multiagent systems by leveraging a hierarchical algorithm. The proposed method effectively transforms the consensus challenge into a tracking problem between agents in adjacent layers, offering a solution to improve the performance of hierarchical consensus algorithms.

Could you provide a more detailed explanation of how the hierarchical algorithm transforms the consensus problem into a tracking problem between agents in adjacent layers? What specific transformations or mappings are involved in this process?

How are the topological weights incorporated into the hierarchical rules, and what is their practical impact on the partition results? Are there specific scenarios or examples where these weights significantly improve the performance compared to traditional hierarchical algorithms?

Could you elaborate on the design and mathematical formulation of the shifting function? How does it interact with the conventional prescribed performance control approach to eliminate the feasibility condition?

---

### Official Review · Reviewer_6JwR · 2024-08-21
**Review Results**

**Rating:** 5
**Confidence:** 4

**Review:**

This paper presents a new hierarchical algorithm to solve the consensus problem in multi-agent systems by simplifying it to a tracking issue. It enhances previous methods by incorporating the significance of topological weights and removing certain constraints.

Could you elaborate on the advantages of the proposed hierarchical algorithm compared to traditional algorithms?

Please provide the specific form of the shift function, and how it eliminates the feasibility conditions of traditional PPC approach?

Could you explain the meaning of 'Guaranteed-Performance'? What is the difference between it and traditional PPC approach?

---

### Official Review · Reviewer_a3fF · 2024-08-21
**manuscript rejection**

**Rating:** 3
**Confidence:** 4

**Review:**

This paper lacks the necessary discussion about the advantages of this work. Hence, the reviewer suggests rejecting this manuscript.

---

### Decision · Program_Chairs · 2024-09-08

Reject